# GR-AttNet: Robotic grasping with lightweight spatial attention mechanism

**Shengbang Zhou**[1]*, **Zhongsheng Liu**[1], **Chuanqi Li**[1], **Dong Chen**[1], **Shutian Liu**[2,3], **Yadong Liu**[1]

1 Guangxi Key Laboratory of Functional Information Materials and Intelligent Information Processing, Nanning Normal University, Nanning, China, 2 Guangxi Geographical Indication Crops Research Center of Big Data Mining and Experimental Engineering Technology, Nanning Normal University, Nanning, China, 3 Guangxi Key Laboratory of Earth Surface Processes and Intelligent Simulation, Nanning Normal University, Nanning, China

* zhoushengbang@nnnu.edu.cn

## Abstract

Robotic grasping is crucial in manufacturing, logistics, and service robotics, but existing methods struggle with object occlusion and complex arrangements in cluttered scenes. We propose the Generative Residual Attention Network (GR-AttNet), based on the Generative Residual Convolutional Neural Network (GR-CNN). The spatial attention mechanism enhances its adaptability to clutter, and architectural optimization maintains high accuracy while reducing parameters and boosting efficiency. Experiments show GR-AttNet achieves 98.1% accuracy on the Cornell Grasping Dataset, 94.9% on the Jacquard Dataset, with processing speeds of 148 ms and 163 ms respectively, outperforming several state-of-the-art methods. With only 2.8 million parameters, it surpasses various existing models. Simulations further validate over 50% success rates across different scenario complexities. However, challenges remain in handling small object missed detection and severe occlusion cases. GR-AttNet offers a novel, highly practical solution for robotic grasping tasks.

## Introduction

Robotic grasping is widely regarded as one of the most fundamental and application-oriented topics in robotics, with its underlying technologies deeply embedded in industrial assembly, logistics sorting, and service robotics [1]. In recent years, with continuous technological advancements, robotic grasping is rapidly developing towards generalization and intelligence. How to reliably grasp objects of diverse materials, geometries, and poses amidst cluttered and unstructured environments has emerged as a pivotal challenge in modern robotics research [2]. Over the past half-century, two paradigms—analytical and data-driven—have been pursued [3]. Analytical methods primarily address grasping detection through the geometry, physical models, kinematics, or mechanics of

**Data availability statement:** All relevant data for this study are publicly available from the Zenodo (https://doi.org/10.5281/zenodo.17356618) and GitHub repositories (https://github.com/zsliu0304/GR-AttNet).

**Funding:** This study was financially supported by the Guangxi Natural Science Foundation in the form of a grant (2025GXNSFBA069254) received by SZ, ZL, CL, DC, SL, and YL. This study was also financially supported by the Guangxi Key Technologies R&D Program in the form of a grant (GUIKEAB24010143) received by SZ and DC. The funders had no role in study design, data collection and analysis, decision to publish, or preparation of the manuscript.

**Competing interests:** The authors have declared that no competing interests exist.

objects. However, modeling errors and sensor noise are inevitable in the real world, rendering even the most elegant analytical solutions unreliable. Data-driven approaches have therefore emerged: rather than pursuing a perfect physical model, the robot "imitates" grasping strategies from large-scale human demonstrations or self-supervised trial-and-error [4]. These approaches are further divided into "known-object" and "unknown-object" grasping. For known objects, Morales et al. [5] and Miller et al. [6] assume that a 3-D model is available and generate grasp candidates by densely sampling the object surface. The method proposed by Fu et al. [7] involves mimicking human grasping postures and iteratively improving the generated grasping poses through repeated trials. Since most objects in daily life do not have digital models, "unknown-object" grasping has become the central focus of recent research. Morrison's GG-CNN [8] (Generative Grasping Convolutional Neural Network) was the first to map a depth image directly to pixel-wise grasp quality, angle, and width via a fully convolutional network, achieving 73% accuracy on the Cornell dataset. Kumra et al. [9] later introduced RGB-D multi-modal fusion and residual skip connections in GR-CNN (Generative Residual Convolutional Neural Network), pushing accuracy to 97.7% and establishing a new lightweight baseline. Yet being the baseline does not imply being the end-point. When we deploy GR-CNN in realistic simulation, we observe that the predicted grasp poses drift significantly whenever the desktop contains multiple objects or cluttered backgrounds. The root cause is that GR-CNN's residual backbone treats all pixels equally and lacks an explicit mechanism to highlight targets while suppressing disturbances in complex scenes. To address this, we propose GR-AttNet (Generative Residual Attention Network), which retains real-time speed, confines the parameter count to 2.8M, and markedly improves generalization to small objects, severe background clutter, and novel categories.

The main contributions of this paper can be summarized as follows:

1) Building upon GR-CNN, we propose GR-AttNet -- an architecturally optimized algorithm that addresses the prevalent issue where high-performance models typically require millions of parameters to support complex feature extraction and decision-making. Through network architecture refinement and integration of lightweight attention modules, GR-AttNet achieves significant performance improvements with 2.8M parameters. This design not only resolves low success rates in complex grasping scenarios but also mitigates the high computational burden of traditional models, maintaining high precision while enhancing real-time capability and deployment adaptability.

2) We propose a novel lightweight attention mechanism as one of GR-AttNet's core innovations. This mechanism automatically focuses on critical regions while suppressing irrelevant information in complex inputs. By integrating attention modules into GR-AttNet, the system achieves more precise target localization and significantly reduces interference from cluttered backgrounds during grasp detection.

## Related work

Looking back at the evolution of grasp detection, early work relied heavily on hand-crafted features. Saxena et al. (2008) [10] was the first to replace analytic force-closure with supervised learning: a 32×32 sliding window on synthetic RGB images trained an MLP to predict grasping points, yet the absence of depth caused shadows to be misclassified as object boundaries, yielding a sharp accuracy drop in cluttered scenes. Subsequently, Jiang et al. (2011) [11] introduced a seven-dimensional grasp rectangle and fused RGB-D features, raising accuracy from Saxena's 60.5% to 73.7% on the Cornell Grasping Dataset, but the high computational cost limited practicality.

With the rise of deep learning, the introduction of convolutional networks greatly enhanced feature extraction. Lenz et al. (2015) [12] proposed a two-stage cascaded CNN: a shallow network first generates hundreds of candidates, then a deeper CNN selects the best one, yet the serial pipeline still fell short of millisecond closed-loop control. Nevertheless, the cascaded CNN framework proposed by Lenz not only demonstrated the advantages of staged processing in improving detection accuracy but also provided ideas for further optimizing the network structure in subsequent research. Xia Jing et al. (2018) [13] combined the R-FCN (Region-based Fully Convolutional Network) and AngleNet modules. The method they proposed achieves significant breakthroughs in improving detection accuracy. However, the generalization ability of this method is limited in complex scenarios. During the research process, researchers found that although the two-stage processing mechanism improves detection accuracy, it also increases computational complexity and processing time. This, to some extent, limits its application in scenarios with high real-time requirements.

It was not until Morrison's GG-CNN [8] abandoned proposals and regressed pixel-wise grasp parameters with a fully convolutional network that inference fell below 19 ms, enabling 50 fps real-time grasping. Its 73% accuracy, however, still falls short of industrial demands. In 2020, Kumra et al. [14] added early RGB-D fusion and residual bottlenecks to GG-CNN, boosting accuracy on the Cornell Grasping Dataset to 97.7% with only 1.9 M parameters. However, GR-CNN struggles to devise effective grasping strategies in cluttered scenes: when two objects are in close proximity, its limited receptive field causes the network to predict a single grasp pose instead of the two distinct grasps that are actually required. Li et al. [15] proposed an RGB-D image fusion-based densely occluded grasping objects detection method by integrating GR-CNN. By separately processing object detection and grasp detection, this method not only effectively enhances the detection accuracy of densely occluded objects but also significantly reduces the workload of data annotation. However, there is still room for improvement in terms of generalization ability and real-time performance in complex scenarios. At this time, attention mechanisms, particularly the Transformer architecture [16], had evolved rapidly and demonstrated success across numerous domains. Lu et al. [17] inserted the SENet (Squeeze-and-Excitation Network) channel-attention module in parallel with a spatial pyramid into GR-CNN, raising the accuracy on the small-target subset from 92.3% to 97.0%. However, when Lu introduced channel-attention module and a spatial pyramid, it led to an increase in the complexity of their network. Specifically, while the application of multi-scale pooling on single feature maps improved accuracy, it led to a sharp decrease in frame rate. Wang et al. [18] introduced a cross-modal fusion grasp detection method based on a hybrid Transformer-CNN architecture. By employing a cross-modal feature interaction fusion module and a dual-stream parallel Transformer-CNN hybrid network architecture, the method significantly enhanced the accuracy and robustness of grasp detection. However, when dealing with scenes where objects are densely packed and closely arranged, the model's detection accuracy tends to decline. This is attributed to the model's difficulty in precisely delineating the boundaries and grasp regions of each object amidst such complex scenarios. Besides, the CBAM module [19], which couples channel and spatial attention, boosts detection accuracy for complex, occluded objects (e.g., wrenches) from 78% to 86% (+8%) in multi-object heaps, effectively suppressing background clutter. Xiong et al. [20] introduced HMT-Grasp, a robotic grasp detection method based on a hybrid Mamba-Transformer architecture. By leveraging a state space model (SSM) to further integrate local and global features, this approach effectively balances fine-grained local details with broader contextual information, thereby enhancing the overall feature representation. While HMT-Grasp achieved a remarkable accuracy of 99% on the Cornell Grasping

                                                                3 / 16

dataset, its inference speed is relatively slow when dealing with complex real-world scenarios, posing challenges to meeting the real-time requirements of practical applications.

In summary, there seems to be a fundamental trade-off in robotic grasp detection methods. Traditional CNN-based approaches can achieve fast inference speeds, with the best performance reaching 30 milliseconds. However, these methods often struggle with accuracy, especially in complex environments where recognition rates can drop significantly. In contrast, methods incorporating attention mechanisms can achieve higher accuracy on datasets but tend to have slower inference speeds, which can limit their practical application in real-time scenarios. The proposed GR-AttNet effectively balances real-time performance and accuracy. By incorporating an attention mechanism, it better handles complex scenarios while achieving processing speeds in the 150ms range, thus better meeting real-time requirements.

## Materials and methods

### Grasp point definition

In this experiment, we consider the robot grasping task as predicting the poses of unknown objects from n-channel images. Unlike the seven-dimensional grasp representation used for spatial grasp, planar grasping typically employs a five-dimensional grasp representation. This experiment adopts a grasp representation method similar to that proposed by Morrison et al. [21]. Specifically, the grasping pose $G_r$ in the robot coordinate system frame is represented as shown in (1).

$$G_r = (P, \Phi_r, W_r, Q) \tag{1}$$

Where $P = (x, y, z)$ is the center position of the end-effector, $\Phi_r$ indicates the angle of rotation around the z-axis. $W_r$ represents the opening width of the end-effector. And $Q$ is the grasp quality score, which is used to evaluate the probability of success for the generated grasp pose. In detail, this score ranges from 0 to 1, with higher values corresponding to a greater probability of success.

Suppose we are conducting grasping detection on a n-channel image $I = \mathbb{R}^{n \times h \times w}$ with height $h$ and width $w$. The grasping pose $G_i$ generated in the image coordinate system is represented by (2).

$$G_i = (\overline{P}, \Phi_i, W_i, Q) \tag{2}$$

Where $\overline{P} = (x, y)$ presents the center point of the grasping in the image coordinate, $\Phi_i$ indicates the angle of rotation in the image coordinate system, which satisfies $\Phi_i \in [-\frac{\pi}{2}, \frac{\pi}{2}]$. $W_i$ represents the width required for grasping, which satisfies $W_i \in [0, W_{max}]$ and $Q$ is the scalar as in (1). It should be noted that $W_{max}$ represents the maximum width that the end-effector can grasp.

To execute the grasping pose obtained in the image space on the robot, the following transformation is required to map the image coordinate system to the robot coordinate system, as shown in (3).

$$G_r = T_w^R[T_c^w(G_i)] \tag{3}$$

$T_c^w$ is the transformation matrix that converts coordinates from the image coordinate system to the camera coordinate system based on the intrinsic parameters of the camera. Similarly, $T_w^R$ maps the camera coordinate system to the robot coordinate system.

### GR-AttNet architecture

GR-CNN achieves an end-to-end mapping from input n-channel images to grasping poses by integrating a generative architecture [22]. This network can directly learn grasping poses from depth images without the need for pre-sampling of

objects. Specifically, GR-CNN first crops the depth images into squares and then resizes them to a resolution of 224 × 224 pixels for input to the network. After passing through the GR-CNN network, it outputs a grasping box including the grasping center position, rotation angle, opening width, and grasping quality score. In this structure, less attention is paid to the background, which makes it difficult to accurately distinguish the target object from the background area.

In response to the above-mentioned problems, inspired by GR-CNN, we have designed a novel network architecture -- GR-AttNet. As shown in Fig 1, the proposed GR-AttNet model integrates n-channel images and depth map as the input. After passing through three convolutional layers, the generated feature maps are processed by the newly proposed attention mechanism module in this experiment and combined with the original feature maps. The combined feature maps are then input into three residual layers. To preserve the spatial features of the image and to ensure the accuracy of the coordinate transformations between the image coordinate system and the robot coordinate system, we up-sample the image to its original size by using a convolution transpose operation. Finally, two images are generated: one contains the grasping angle and width required by the robotic arm in the real world, and the other is used to evaluate the quality of the generated grasping box.

To ensure that the network can effectively perform feature extraction and processing, GR-CNN convolves the images into a size of 56 × 56, enabling the network to learn more abundant features while avoiding excessive computational load. In this experiment, we will continue to use this operation for image feature extraction. After convolution, the features are passed into the attention mechanism module that we proposed. The specific structure of this module will be detailed in the next section. Then, combined feature maps are fed into three residual networks, each of which consists of two convolutional layers, two normalization layers and an activation layer. The convolutional layer is responsible for feature extraction,

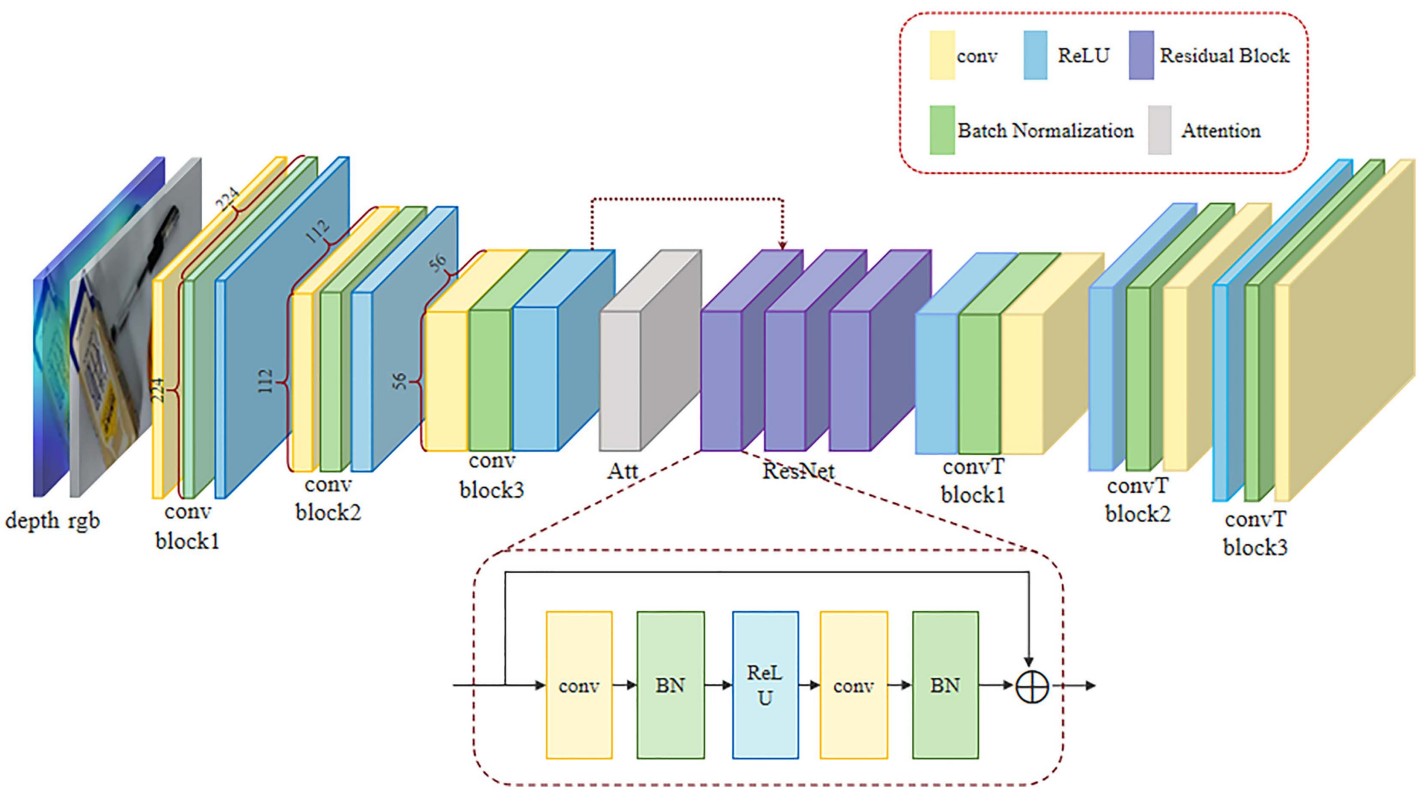

**Fig 1. GR-AttNet architecture.**

the normalization layer is used to stabilize training and accelerate convergence, and the activation layer introduces non-linear characteristics to enhance the model's expressive power. This structural design is simple and efficient, capable of effectively processing input features and providing high-quality feature representations for subsequent tasks.

As we know, residual networks can effectively address the degradation problem in training deep neural networks, that is, the phenomenon where the training and validation errors increase with the increase of network depth. The residual network enables the network to learn the residual function $F(x) = H(x) - x$ instead of directly learning the objective function $H(x)$. This approach simplifies the optimization process. However, when the number of residual networks exceeds a certain limit, it can lead to problems such as vanishing gradients and dimensional errors, resulting in a decrease in accuracy. Given that the network achieves effective feature fusion through our attention mechanism, the residual block can fully capture image features with only a three-layer structure. Therefore, to reduce the number of model parameters, the residual network in this experiment is designed to have three layers.

The proposed network contains only 2.8 M parameters, compared to 35.2 M of the Oriented-Anchor FCN in [23] and 46.7 M of the ConvNet-Grasp in [24], making it significantly more lightweight. Therefore, when compared to similar grasping prediction techniques that utilize complex architectures and involve millions of parameters, this network demonstrates significant advantages. It achieves lower computational costs and faster running speeds, making it more computationally efficient overall.

**Attention block**

The spatial attention mechanism assigns varying weights to different positions within the input image based on their significance, thereby enhancing the model's focus on and extraction of key information [25]. Additionally, the attention mechanism adaptively adjusts the weights of channels, which strengthens GR-AttNet's ability to learn grasping-related features [26]. This adaptive weighting effectively suppresses irrelevant feature information, reduces prediction bias, and consequently improves the model's robustness and the accuracy of grasping detection [27].

Inspired by the structure of SK-Net [28], this experiment designs the attention mechanism module as shown in Fig 2. This module introduces multiple parallel convolutional kernel branches with different receptive fields to learn the weights of feature maps at different scales, enabling the GR-AttNet network to select more appropriate multi-scale feature representations. Assuming the input feature map **V** has dimensions $h \times w \times c$, where $h$ and $w$ are the spatial height and width of the feature map, and $c$ is the number of channels. The specific representation is given by the following equation (4).

$$\mathbf{V} = [v^{(1,1)}, v^{(1,2)}, ..., v^{(i,j)}, ..., v^{(h,w)}] \tag{4}$$

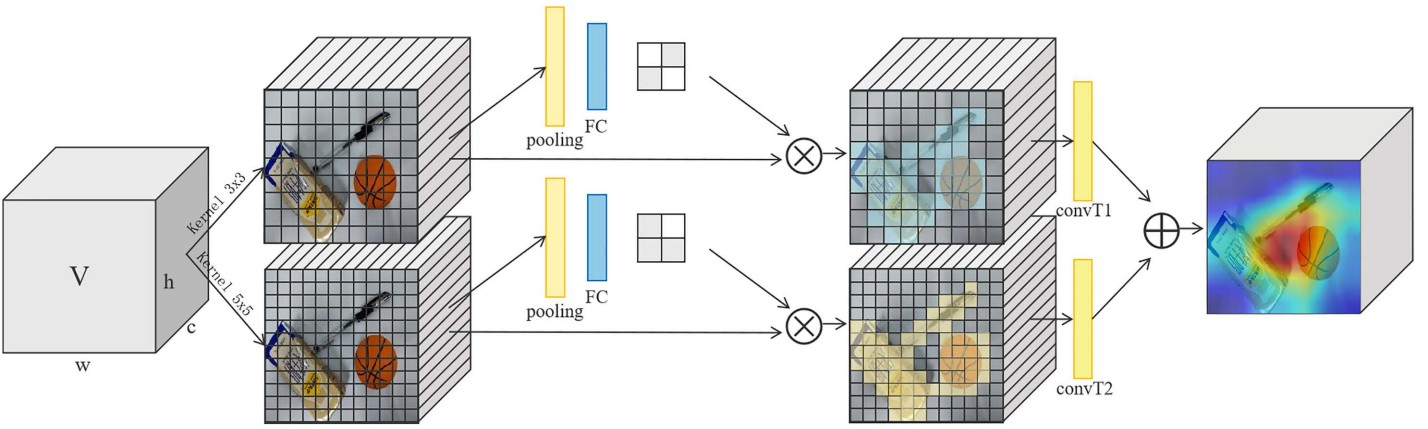

**Fig 2. Structure of attention mechanism block.**

Where $v^{(i,j)} \in \mathbb{R}^{(1 \times 1 \times c)}$, $i \in [1, h]$, $j \in [1, w]$. The input feature map V is first convolved with two kernels of size 3×3 and 7×7. The selection of these two sizes of convolution kernels will be discussed in detail in Results and Discussion. The smaller kernel (3×3) is suitable for extracting detailed information, while the larger kernel (7×7) covers a broader receptive field to capture more extensive contextual information. To facilitate subsequent calculations, this study employs the "same" padding method, ensuring that the size of the feature map after convolution remains consistent with the original feature map. The specific convolution processes are denoted as $\mathbf{U} = K_{sq} * \mathbf{V}$ and $\mathbf{U}' = K'_{sq} * \mathbf{V}$, where $K_{sq} \in \mathbb{R}^{(3 \times 3 \times c)}$, $K'_{sq} \in \mathbb{R}^{(7 \times 7 \times c)}$, $\mathbf{U} \in \mathbb{R}^{(H,W)}$, $\mathbf{U}' \in \mathbb{R}^{(H,W)}$. Afterwards, average pooling is applied to each channel of the two convolved feature maps, resulting in the global spatial features for each channel, forming two tensors of size $h \times w \times 1$. Channel-wise average pooling effectively captures the global information of each channel without increasing the model's parameter count. Subsequently, the pooled feature maps are passed through a fully connected layer that includes a Sigmoid function, which performs a nonlinear transformation on the feature vectors. The Sigmoid function normalizes the attention scores into a probability distribution, ensuring that the attention weights for each spatial location fall between 0 and 1. This process yields the weights for each spatial position.

Then, the generated weights are multiplied element-wise with the original feature map. As shown in (5) and (6), this adjusts the contributions of each channel in the original feature map, enabling the model to focus more on important features while suppressing less important ones.

$$\widehat{V} = [\sigma(u_{(1,1)})v^{(1,1)}, \sigma(u_{(1,2)})v^{(1,2)}, ..., \sigma(u_{(i,j)})v^{(i,j)}, ..., \sigma(u_{(h,w)})v^{(h,w)}] \tag{5}$$

$$\widehat{V}' = [\sigma(u'_{(1,1)})v^{(1,1)}, \sigma(u'_{(1,2)})v^{(1,2)}, .., \sigma(u'_{(i,j)})v^{(i,j)}, .., \sigma(u'_{(h,w)})v^{(h,w)}] \tag{6}$$

Finally, the two feature maps $\widehat{V}$ and $\widehat{V}'$ are fused at the pixel level as shown in (7).

$$\widetilde{V} = \widehat{V}' + \widehat{V} \tag{7}$$

The attention mechanism learns the importance of different channels in the input feature map and adjusts it accordingly. thereby enhancing the model's ability to capture key information. All in all, the introduction of the attention mechanism enhances the model's ability to capture key information.

### Training

**Evaluation standards.** To objectively compare the performance of different grasping detection models, this study introduces the evaluation standard proposed by Jiang et al. [11] in 2011 to evaluate the accuracy of the grasping box. According to this rectangular metric, a grasping prediction is deemed correct if it meets the following conditions.

1) The difference between the predicted grasping angle and the actual grasping angle is within 30°.

2) The Jaccard index between the predicted grasping rectangle and the actual grasping rectangle is greater than 0.25. The definition of the Jaccard index is as follows:

$$J(G_P, G_t) = \left| \frac{G_P \cap G_t}{G_P \cup G_t} \right| > 0.25 \tag{8}$$

Where $G_P$ represents the predicted capture rectangle, $G_t$ represents the true grasping rectangle.

**Grasp datasets.** In the field of robotic planar grasping, the number of publicly available datasets is limited. Table 1 summarizes several representative datasets, each with its own merits and demerits. The Cornell Grasping dataset is

**Table 1. Different types Datasets.**

| Datasets Name | Images | Grasps | Source | Scene | Method |
|---|---|---|---|---|---|
| Cornell [12] | 1,035 | 8,019 | Real | Single object | grasping box |
| Jacquard [29] | 54,000 | 1.1M | Synthesis | Single object | grasping box |
| Grasp-Net [30] | 97,000 | 1.2B | Real | Multiple objects (1~10) | 6-DOF pose |
| Dex-Net [31] | 6.7M | 6.7M | Synthesis | Single object | grasping box |
| VR-Grasping-101 [32] | 10,000 | 4.8M | Synthesis | Single object | 6-DOF pose |
| Multi-Object [33] | 96 | 2,904 | Real | Multiple objects (4~5) | grasping box |

the most common benchmark dataset, containing 1,035 RGB-D images and 8,019 grasping annotations, covering 240 objects. However, its scenes are relatively simple, with only a single object in each image, making it difficult to meet the grasping requirements in complex scenarios. The Jacquard dataset is larger in scale, with 54,000 RGB-D images and 1.1 million grasping annotations, covering 11,619 objects. Its synthetic data approach provides rich diversity, but there is a certain gap from real-world scenarios. Grasp-Net further expands the scale and complexity, with 97,000 RGB-D images and 1.2 billion grasping annotations, covering 88 objects, and supports multi-object scenarios (1–10 objects) and 6-DOF grasping pose annotations. However, the processing and training costs of large-scale data are relatively high. DexNet focuses on depth images, containing 6.7 million depth images and 6.7 million grasping annotations, covering 1,500 objects. Its synthetic data is helpful for studying the robustness of grasping algorithms, but it only relies on depth information and may be limited in complex visual scenarios. The VR-Grasping-101 dataset is generated through virtual reality technology, containing 10,000 RGB-D images and 4.8 million grasping annotations, covering 101 objects. The Multi-object dataset is based on the annotation method of the Cornell Grasping Dataset, placing multiple objects in the same single scene for recognition, containing 96 RGB-D images and 2,904 grasping annotations. It makes up for the deficiency of the Cornell Grasping Dataset in complex scenarios and provides an experimental basis for multi-object grasping research.

Firstly, we focus on the testing of individual objects. To ensure that the objects closely resemble real-world scenarios, the Cornell and Jacquard datasets were selected as experimental subjects. Additionally, to demonstrate the method's superiority in cluttered environments, the Multi-Object dataset was also tested. These three datasets contain objects of different shapes, sizes, materials, and textures, which can comprehensively evaluate the performance of the network under various conditions.

**Simulation environment setup.** To verify the superiority of the proposed network while keeping costs low, all simulation experiments were conducted using PyBullet—an open-source Python module built on the Bullet physics engine—running under Windows. Complex physical phenomena (e.g., collision detection, friction) can be effectively simulated through PyBullet's rigid-body dynamics engine, which also exhibits high computational efficiency in real-time scenarios.

The robotic arm model selected is the UR5, which features six rotary joints with motion ranges of [0, 2π] for each joint. Its end-effector is a two-finger gripper. The UR5 is characterized by high precision and sensitivity, and it offers high control efficiency. The comprehensive ecosystem of the UR5 enables it to be easily integrated with other technologies to achieve object detection and complex grasping tasks. The simulation process is shown in Fig 3 and mainly consists of the following four steps:

1] Environment Setup: In PyBullet, we construct a simulation environment that includes a robotic arm and a variety of objects. We set all objects as rigid bodies, including many irregular objects such as keychains and dolls. We randomly place these objects on the ground, while the position of the robotic arm is fixed.

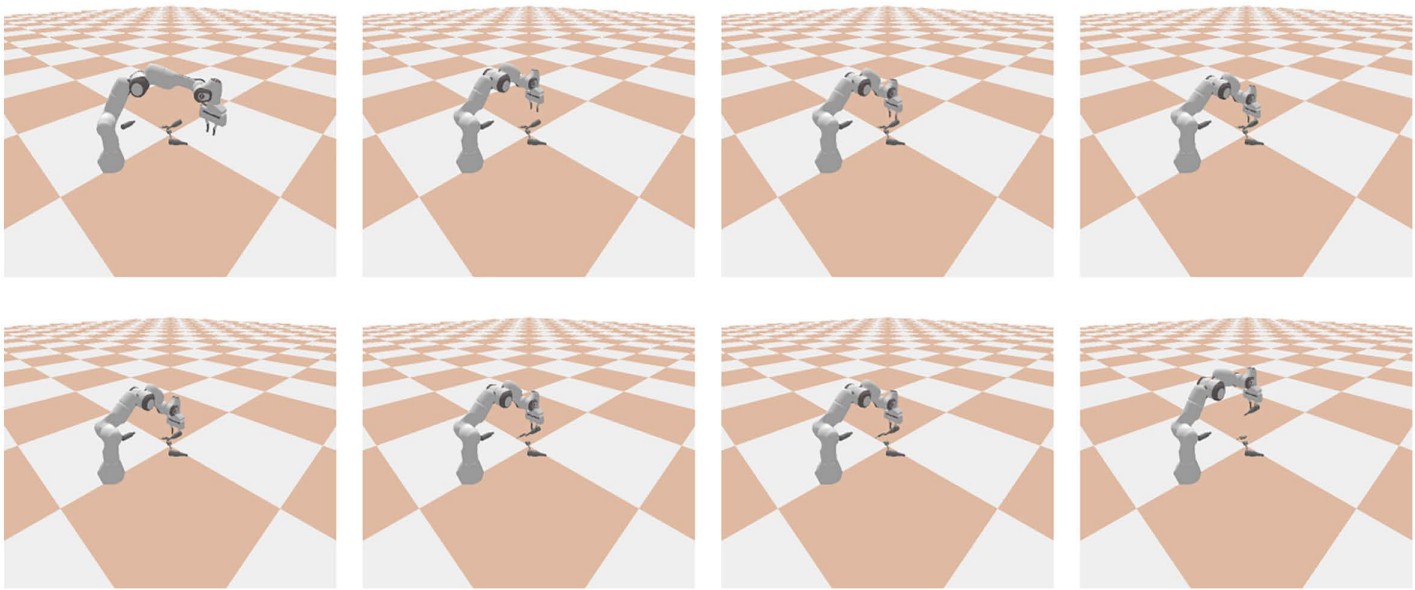

**Fig 3. Simulation environment and specific capture process.** Starting from the first image on the left in the first row and moving to the right, it illustrates a specific grasping process step by step.

2] Visual Perception: Depth pixel information of target objects was acquired through an RGB-D camera (Fig 4).

3] Grasp Prediction: Object grasp poses were predicted using the GR-AttNet network.

4] Execution & Evaluation: The best redicted grasping box is mapped to the simulated robotic arm to perform the grasp. If an object is successfully grasped, it is removed from the environment. A maximum of 5 grasp attempts were allocated per object. If all attempts failed, the system reinitialized visual perception and generated new grasp proposals (returning to Step 2).

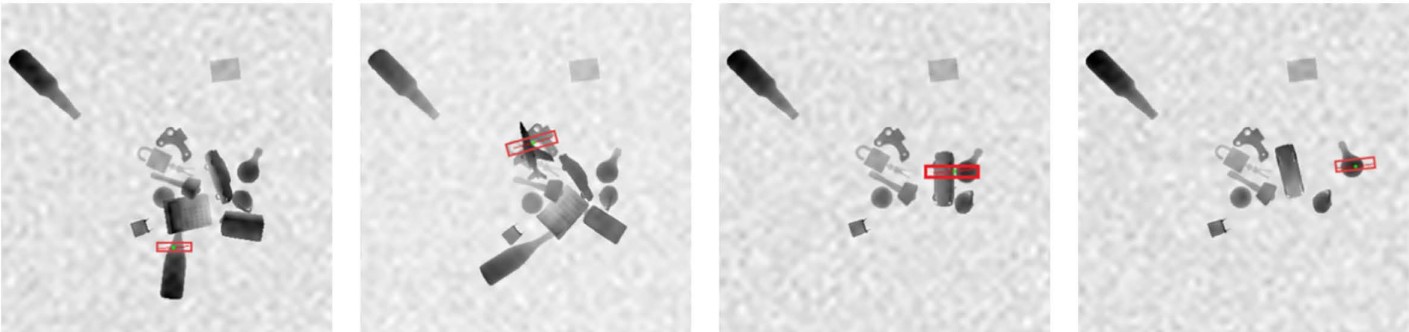

**Fig 4. Visual Perception.** The camera captures information and generates a grasping box for one object at a time. After the robotic arm successfully grasps the object, the successfully grasped object is deleted from the environment, and the process continues in this manner.

# Results and discussion

## Training results

### a) Attention mechanism results

To thoroughly investigate the impact of kernel size in the proposed attention module on network performance, a series of experiments were designed in this paper. By selecting convolutional kernels of different sizes and applying them to the constructed network architecture, detailed records and analyses were conducted during the training and testing processes of the network. Each convolution kernel combination is tested through five independent experiments, and the results are averaged. The experimental results are shown in Table 2, which indicate that changes in kernel size have a significant influence on the performance of the network structure. Convolutional kernels of different sizes exhibit varying performance in terms of feature extraction, computational complexity, and model generalization ability. Specifically, in the network architecture proposed in this paper, convolutional kernels of size 3×3 and 7×7 achieved the highest accuracy, reaching 98.1% on the Cornell Grasping Dataset, with only 2.8M parameters. On the Jacquard dataset, convolutional kernels of size 5×5 and 7×7 achieved the highest accuracy of 95.1%, but the number of parameters also increased to 3.1M.

Through in-depth analysis, we have determined that the combination of 3×3 and 7×7 convolutional kernels achieves the highest accuracy and the lowest mean squared error. This success is primarily attributed to the ability of these kernels to extract features at different scales. Specifically, the 3×3 convolutional kernel excels at capturing local details in images, such as edges and textures. These details are essential for accurately identifying the fine-grained structure of objects. Conversely, the 7×7 convolutional kernel is adept at extracting broader contextual information, including the contours and overall shapes of objects [34]. This capability enables the model to better understand the spatial relationships and global structure within an image. By integrating these two kernels, the model gains a comprehensive understanding of both local details and global context.

### b) GR-AttNet training results

Due to the relatively small size of the Cornell dataset, five-fold cross-validation is adopted in this paper. Table 3 shows the accuracy rates of different grasping detection models on the Cornell dataset. The GR-AttNet achieves an accuracy rate of 98.1%, which is higher than the baseline GR-CNN's 97.7%. In addition, compared with other large-scale networks, GR-AttNet has only 2.8M parameters. Furthermore, due to the linear topology of GR-AttNet, which incorporates the attention module only after the completion of convolution, the overall complexity remains very low. This enables convenient deployment on edge devices. Table 4 presents the accuracy rates of various models on the Jacquard dataset. GR-AttNet achieves an accuracy rate of 94.9%, which is also higher than the baseline GR-CNN's 94.6%. Although the baseline method is theoretically capable of achieving an inference speed of 20 ms, we found that its actual running speed is about 200 ms in our tests. The root cause of this discrepancy lies in the difference in timing methodologies: the 20 ms reported by Kumra et al.

**Table 2. The effect of kernel size on the experiment.**

| Size | | | Parameters | Accuracy(%) | |
|------|------|------|------------|-------------|---------|
| 3×3 | 5×5 | 7×7 | | Cornell | Jacquard |
| √ | | | 2.0M | 92.4±0.3 | 90.8±0.4 |
| | √ | | 2.3M | 94.5±0.3 | 91.3±0.3 |
| | | √ | 2.7M | 95.5±0.2 | 92.9±0.3 |
| √ | √ | | 2.4M | 95.4±0.2 | 94.2±0.3 |
| √ | | √ | 2.8M | 98.1±0.2 | 94.9±0.2 |
| | √ | √ | 3.1M | 96.3±0.3 | 95.1±0.3 |
| √ | √ | √ | 3.3M | 96.7±0.2 | 94.3±0.3 |

† The mean±SE is derived from five independent training runs.

**Table 3. Performance of different models on Cornell dataset.**

| Author | Network structure | Accuracy(%) | Parameters | Speed(ms) |
|---|---|---|---|---|
| Jiang [11] | Fast Search | 60.5 | \ | 5000 |
| Morrison [8] | GG-CNN | 73 | 62000 | 19 |
| Lenz [12] | SAE | 73.9 | \ | 1350 |
| Redmon [38] | AlexNet | 88 | ~6.1M | 76 |
| Kumra [9] | GR-CNN | 97.7 | 1.9M | 20 |
| Lou [39] | GR-CNN+CBAM | \ | 6.1M | \ |
| ours | GR-AttNet | 98.1 | 2.8M | 148 |

**Table 4. Performance of different models on Jacquard dataset.**

| Author | Network structure | Accuracy(%) | Speed(ms) |
|---|---|---|---|
| Depierre [29] | Jacquard | 74.2 | \ |
| Morrison [8] | GG-CNN2 | 84 | 20 |
| Zhou [23] | FCGN | 91.8 | 117 |
| Kumra [9] | GR-CNN | 94.6 | 20 |
| Yu [40] | PTGNet | 94.8 | 42.5 |
| ours | GR-AttNet | 94.9 | 163 |

refers to the GPU kernel time, whereas in this study, after cascading the attention module with the network linearly, the measurement encompasses the entire Python inference pipeline (including post-processing steps such as angle conversion). Consequently, the actual measured result is 200 ms, and the one-order-of-magnitude difference between the two is within the expected range. In contrast, the method proposed in this paper only takes 148 ms and 163 ms to complete inference in practical testing. As reported in Zeng [35] and Vidovič [36], a 200 ms vision-to-actuation latency is the upper bound for 0.5m/s conveyor picking to keep error < 10 mm. For domestic service robots, Goodrich [37] shows that ≤ 400 ms is perceived as instantaneous by users. Our 148 ms thus meets industrial "pass-line" and comfortably satisfies service-robot timeliness.

In summary, GR-AttNet can perform the planar grasping task well. However, due to the introduction of the attention mechanism, although the model has fewer parameters, it leads to a more complex computational process compared to GR-CNN, requiring more computing resources for processing. Therefore, GR-AttNet is slightly inferior to other high performance models in terms of inference speed. Nevertheless, GR-AttNet demonstrates excellent performance in grasp detection accuracy and generalization across different object geometries, particularly in cluttered or partially occluded scenarios. The attention mechanism enables the model to focus on the most relevant spatial features, enhancing robustness relative to traditional CNN-based methods. This makes GR-AttNet especially suitable for applications where precision and adaptability are highly valued.

To present the research results of this paper more intuitively, the visualization results of the grasping outcomes are shown in Fig 5 RGB represents the original RGB image. Att displays the image effect after processing by the designed attention mechanism module. The target area is effectively highlighted. It is evident from this that the module can precisely focus on the items to be grasped. Grasp is the grasping box generated by the network, accurately marking the grasping position. Q represents the final grasping quality map, which visually reflects the reliability of the grasping operation.

Moreover, we tested the proposed method on the Multi-Cornell dataset [33] and achieved an accuracy of 89.2%, with some of the grasping results shown in Fig 6-8.

These visualized results clearly demonstrate that GR-AttNet is capable of accurately generating grasping boxes, thereby providing precise guidance for subsequent grasping operations.

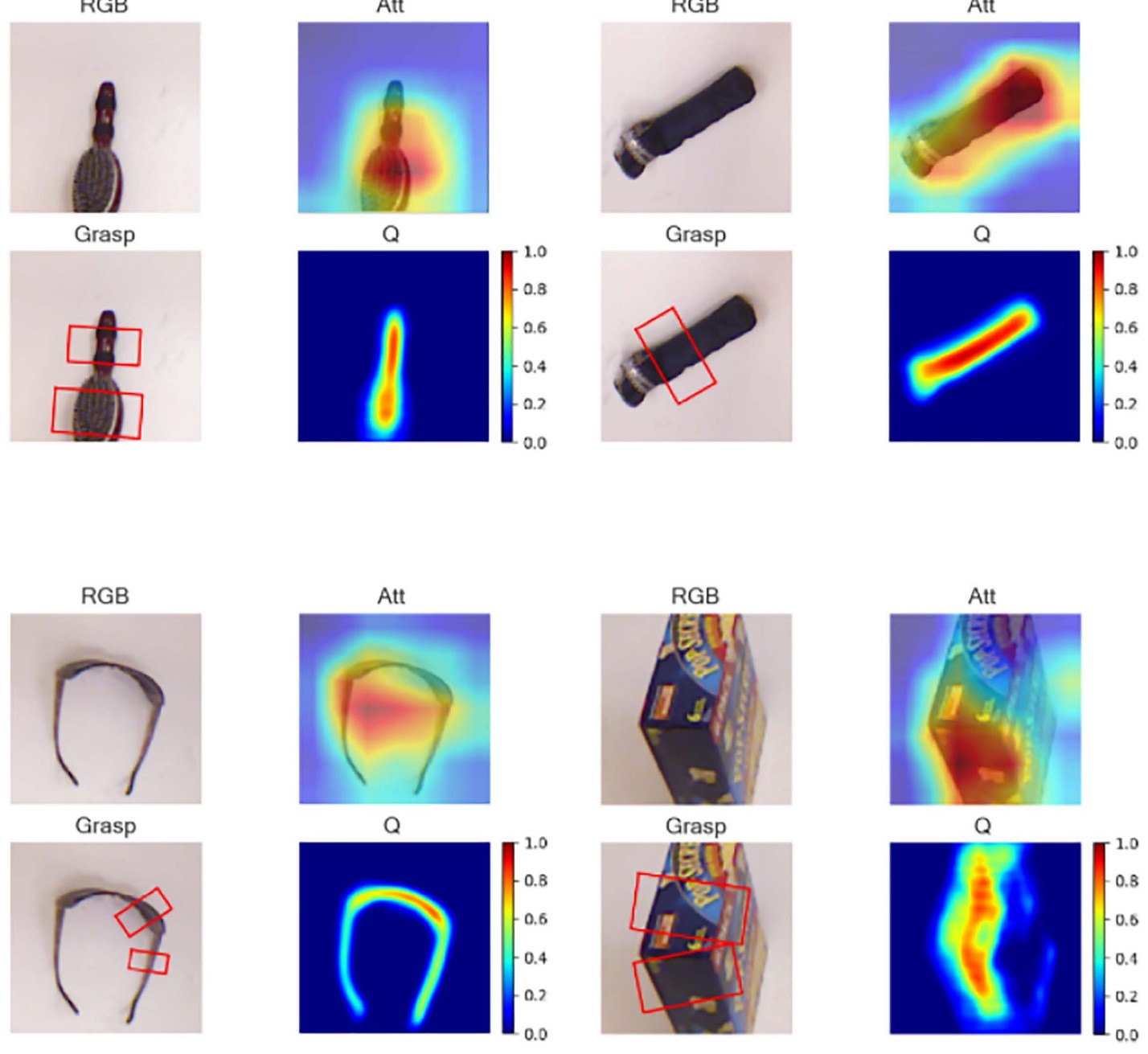

**Fig 5. Visualization of the single results:** RGB represents the original RGB image; Att displays the image effect after processing by the designed attention mechanism module; Grasp is the grasping box generated by the network; Q represents the final grasping quality map.

## Simulation results

In simulation, we conducted 50 trials for each of the four cases of grasping 1, 5, 10 and 15 objects. A trial was declared a failure if the same object could not be grasped within five successive attempts. We recorded the total number of grasp attempts, the types of failure, and the overall success probability; the results are summarized in Table 5.

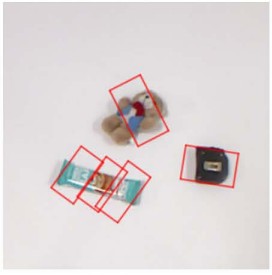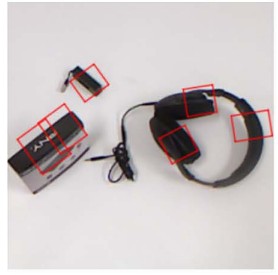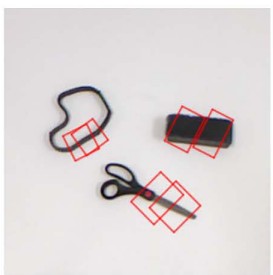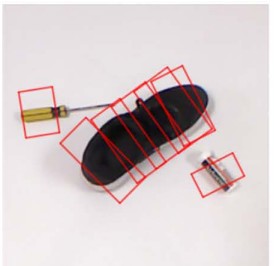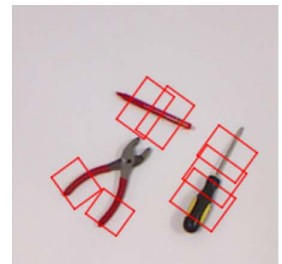

**Fig 6. Visualization of the Multi-Objects results.** The image shows the grasping boxes generated by the GR-Att network, with multiple boxes for each object.

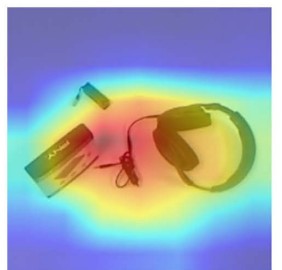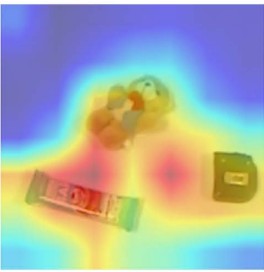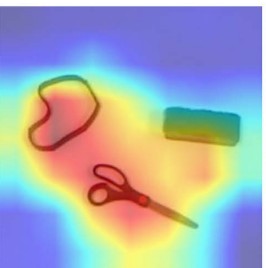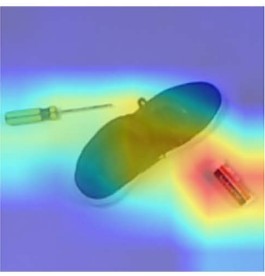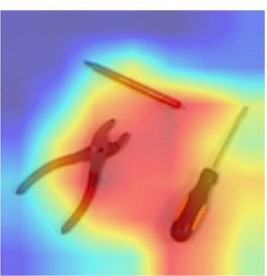

**Fig 7. Attention regions of the Multi-Objects results.** The image shows the attention regions generated by the Att module.

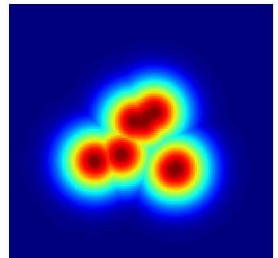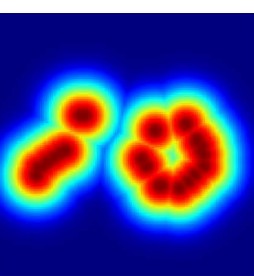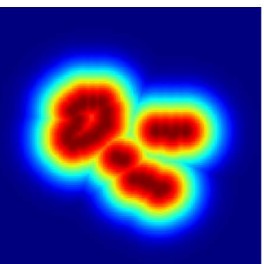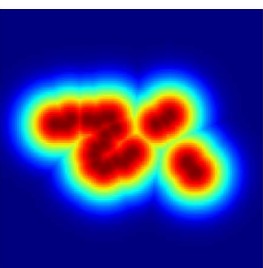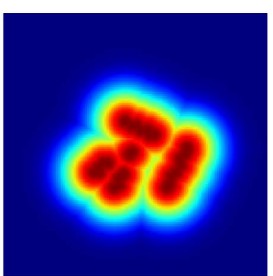

**Fig 8. The grasp quality score of the Multi-Objects.** These images illustrate the grasp quality of multiple objects.

**Table 5. Simulation and grasping results of different kinds of objects.**

| Species | Total number | Number of success | Failure times | | | Accuracy(%) |
|---|---|---|---|---|---|---|
| | | | Slippage times | Occlusion errors | Recognition errors | |
| 1 | 59 | 47 | 2 | 0 | 10 | 79.70% |
| 5 | 318 | 247 | 18 | 10 | 43 | 77.67% |
| 10 | 611 | 412 | 52 | 76 | 71 | 67.43% |
| 15 | 1639 | 701 | 152 | 282 | 504 | 42.77% |

The grasping success rate reaches a relatively high level of 79.7% in single-object scenarios. The term "Slippage times" refers to the number of failures during the grasping process where the gripper comes into contact with other objects, leading to inaccurate recognition and subsequent failure. "Occlusion errors" refer to the number of failures where the relationship between objects in terms of depth is not correctly identified. As a result, the generated grasping box directs the robotic arm to grasp an object that is further back first, leading to failure. "Recognition errors" refer to the number of failures caused by the generation of unreasonable grasping boxes during the grasping process.

To better assess the statistical robustness of the simulation results presented in Table 5, we calculated 95% confidence intervals (CIs) for the success rates across different object quantities using the Wilson score interval method [41]. For single-object grasping, the success rate of 79.7% yields a 95% CI of [68.1%, 87.8%], indicating a relatively reliable performance. However, as the number of objects increases, the CIs widen significantly: for 5, 10, and 15 objects, the 95% CIs are [72.8%, 81.7%], [63.6%, 71.0%], and [40.4%, 45.2%], respectively. According to the Wilson formula estimation, if the number of trials is increased to four times, the half-width of the 95% CI can be reduced to approximately ±5.6%, and further verification will be conducted through supplementary experiments. This trend confirms that the model maintains robust, above-random performance across all complexity levels. Moreover, prevailing high-performance architectures benchmark grasping solely on isolated items: Lou [25] reports accuracy only for a single, pre-selected object; the highest rate, 90%, was achieved on a soda can. And GG-CNN cites 87% success (83/96) in single-object simulations, leaving occlusions and multi-object clutter unaddressed.

Through simulation visualization and further analysis, the following critical issues have been identified. For small-sized objects, although the network can accurately identify their features and generate corresponding grasping poses, the physical grasping process often fails due to object slippage caused by their miniature dimensions. Conversely, for large-sized objects, the network occasionally produces invalid grasping boxes that exceed the maximum grasping range, similarly resulting in failed grasping operations.

Furthermore, the network demonstrates significant limitations in handling occlusions between objects. When the network initially identifies a partially occluded object and generates a grasp pose, the end-effector frequently collides with obstructing objects during approach. This unintended contact displaces the target object, ultimately causing grasp failure.

## Conclusion

This study proposes a multi-modal end-to-end grasping framework -- GR-AttNet. By introducing the attention mechanism and optimizing the network architecture, the system significantly improves grasping detection accuracy and robustness in cluttered scenarios.

The experimental results show that GR-AttNet achieves an accuracy of 98.1% and 94.9% on the Cornell Grasping Dataset and the Jacquard Dataset. The network shows good adaptability in the grasping task of different numbers of objects in the simulated environment, and still achieves a 50% grasping success rate in the face of severe occlusion. However, sometimes there is still the problem of not obtaining the target position in real time, and in the face of antagonistic objects, the network processing results and the actual situation are quite different. In the future, we plan to deploy and validate on actual robotic platforms (such as the UR5 robotic arm) to further test the generalization capability and practical application value of GR-AttNet. Despite these limitations, the proposed GR-AttNet demonstrates robust generalization capabilities and practical feasibility, establishing a valuable reference framework for robotic planar grasping applications in real-world environments.

## Acknowledgments

We gratefully acknowledge the Guangxi Key Laboratory of Functional Information Materials and Intelligent Information Processing for providing experimental facilities and technical support.

# Author contributions

**Data curation:** Shutian Liu.

**Project administration:** Yadong Liu.

**Software:** zhongsheng liu.

**Validation:** Yadong Liu.

**Visualization:** Shutian Liu.

**Writing – original draft:** zhongsheng liu.

**Writing – review & editing:** Shengbang Zhou, zhongsheng liu, Chuanqi Li, Dong Chen, Shutian Liu.

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
