## [Decision Letter · Decision Letter 0]

2 Sep 2025

PONE-D-25-28907GR-AttNet: Robotic Grasping with Lightweight Spatial Attention MechanismPLOS ONE

Dear Dr. Zhou,

Thank you for submitting your manuscript to PLOS ONE. After careful consideration, we feel that it has merit but does not fully meet PLOS ONE’s publication criteria as it currently stands. Therefore, we invite you to submit a revised version of the manuscript that addresses the points raised during the review process.

**Improvements of the results compared with existing works should be clearly presented and discussed. References can be further enhanced while the writing of whole manuscript is suggested to improve.** 

We look forward to receiving your revised manuscript.

Kind regards,

Longhui Qin, Ph.D.

Academic Editor

PLOS ONE

**Journal Requirements:**

1. When submitting your revision, we need you to address these additional requirements. Please ensure that your manuscript meets PLOS ONE's style requirements, including those for file naming. The PLOS ONE style templates can be found at https://journals.plos.org/plosone/s/file?id=wjVg/PLOSOne_formatting_sample_main_body.pdf and https://journals.plos.org/plosone/s/file?id=ba62/PLOSOne_formatting_sample_title_authors_affiliations.pdf 2. Please note that PLOS One has specific guidelines on code sharing for submissions in which author-generated code underpins the findings in the manuscript. In these cases, we expect all author-generated code to be made available without restrictions upon publication of the work. Please review our guidelines at https://journals.plos.org/plosone/s/materials-and-software-sharing#loc-sharing-code and ensure that your code is shared in a way that follows best practice and facilitates reproducibility and reuse. 3. We note that the grant information you provided in the ‘Funding Information’ and ‘Financial Disclosure’ sections do not match.  When you resubmit, please ensure that you provide the correct grant numbers for the awards you received for your study in the ‘Funding Information’ section. 4. Thank you for stating the following financial disclosure: Guangxi Key Technologies R&D Program (No.AB241484046) and Guangxi Science and Technology Program (Project  No. GUIKEAB23075177)   Please state what role the funders took in the study.  If the funders had no role, please state: "The funders had no role in study design, data collection and analysis, decision to publish, or preparation of the manuscript." If this statement is not correct you must amend it as needed. Please include this amended Role of Funder statement in your cover letter; we will change the online submission form on your behalf. 5. Thank you for stating the following in the Acknowledgments Section of your manuscript: This research was supported by Guangxi Key Technologies R&D Program (No.AB241484046) and Guangxi Science and Technology Program (Project  No. GUIKEAB23075177). We also gratefully acknowledge the Guangxi Key Laboratory of Functional Information Materials and Intelligent Information Processing for providing experimental facilities and technical support. We note that you have provided funding information that is not currently declared in your Funding Statement. However, funding information should not appear in the Acknowledgments section or other areas of your manuscript. We will only publish funding information present in the Funding Statement section of the online submission form. Please remove any funding-related text from the manuscript and let us know how you would like to update your Funding Statement. Currently, your Funding Statement reads as follows: Guangxi Key Technologies R&D Program (No.AB241484046) and Guangxi Science and Technology Program (Project  No. GUIKEAB23075177)  Please include your amended statements within your cover letter; we will change the online submission form on your behalf. 6. In the online submission form, you indicated that your data is available only on request from a third party. Please note that your Data Availability Statement is currently missing the contact details for the third party, such as an email address or a link to where data requests can be made. Please update your statement with the missing information. 7. PLOS requires an ORCID iD for the corresponding author in Editorial Manager on papers submitted after December 6th, 2016. Please ensure that you have an ORCID iD and that it is validated in Editorial Manager. To do this, go to ‘Update my Information’ (in the upper left-hand corner of the main menu), and click on the Fetch/Validate link next to the ORCID field. This will take you to the ORCID site and allow you to create a new iD or authenticate a pre-existing iD in Editorial Manager. 8. Please upload a copy of S1 Figure, to which you refer in your text on page 20. If the figure is no longer to be included as part of the submission please remove all reference to it within the text. 9. If the reviewer comments include a recommendation to cite specific previously published works, please review and evaluate these publications to determine whether they are relevant and should be cited. There is no requirement to cite these works unless the editor has indicated otherwise. 

Reviewers' comments:

Reviewer's Responses to Questions

**Comments to the Author**

1. Is the manuscript technically sound, and do the data support the conclusions?

Reviewer #1: Yes

Reviewer #2: No

Reviewer #3: Yes

2. Has the statistical analysis been performed appropriately and rigorously? 

Reviewer #1: No

Reviewer #2: No

Reviewer #3: N/A

3. Have the authors made all data underlying the findings in their manuscript fully available?

Reviewer #1: Yes

Reviewer #2: No

Reviewer #3: No

4. Is the manuscript presented in an intelligible fashion and written in standard English?

Reviewer #1: Yes

Reviewer #2: No

Reviewer #3: Yes

5. Review Comments to the Author

**Reviewer #1:**  The paper proposes GR-AttNet, a particularly new neural network for robotic grasping that combines a generative residual architecture with a lightweight spatial attention mechanism. The key innovations appear to be, firstly, in architectural optimization, which reduces parameters to 2.8M while maintaining accuracy. Secondly, a novel attention module utilizes parallel 3x3 and 7x7 convolutional kernels to capture multi-scale feature fusion. The authors have used standard datasets such as Cornell and Jacquard, and evaluation metrics, e.g., Jiang's rectangle metric, which is suitable for comparability.

However, no statistical significance testing for accuracy improvements, limited discussion on why 3X3 or 7X7 kernels outperformed other combinations, and simulation testing only in PyBullet without real-world validation.

Additionally, the claimed 20ms inference for GR-CNN, which conflicts with their measured 200ms, warrants explanation. Limited exploration of how the attention module generalizes to non-grasping robotics tasks? Despite testing on Jacquard (synthetic) and Cornell (real) datasets, the model struggles with small objects or severe occlusions, as shown in Table 5, indicating limitations in the dataset. Possible to add real-world robot trials to validate simulation results?

Overall, this work significantly advances efficient robotic grasping and provides a practical foundation for real-time applications, although real-world robustness requires further validation.

**Reviewer #2:**  The manuscript proposes GR-AttNet, which is a GR-CNN-style, fully convolutional grasp detector that is augmented with a lightweight spatial attention block containing 3×3 and 7×7 kernels, as well as minor architectural tweaks. The topic, planar robotic grasp detection, is timely and relevant, and the paper's focus on attention modules for robust grasping in clutter is well-motivated.

However, major modifications must be made before publication.

First, some of the references are outdated. More recent works on transformer-based grasping and real-world deployment challenges from 2022–2024 should be discussed in depth. Second, the paper suffers from redundancy and repeated explanations (e.g., the grasp pose definition is repeated almost verbatim). Third, figures are referenced but not always explained in sufficient detail (e.g., the attention module in Fig. 2). Fourth, the "Related Work" section could be improved by critically comparing the limitations of existing methods rather than simply listing them. The simulation results are promising, but they lack statistical robustness (e.g., there are no confidence intervals or discussions of variance across trials). Lastly, there are some minor language and formatting issues that need correction, such as "Error! Reference source not found." and inconsistent terminology.

**Reviewer #3: ** The following are some corrections to be made:

At line 31: There is an error in the reference not being found

At line 40: I believe there would be a reference for Kumra at the location of 'Kumra[ ] introduced the GR-CNN..."

At line 94: add a space between "closedloop" to be "closed loop"

At lines 94-95: It states " enhancing the grasping ability for small objects, with a speed of 23 ms per frame and an accuracy increase to 97%. This is an increase from what or compared to what for small object grasping?

At line 96: the same as the above what is the increasing detection accuracy for complex objects compared to. It states an increase to 86% but compared to what?

At lines 105-108: Why are these lines the same as the lines 101-104 above equation 1? These lines 105-108 need removed.

At line 113 it states "height w and width h", I assume the w and h need switched so it states "height h and width w".

At line 158: It states this experiment only contains 2.8 million parameters. Please add what the standard model has for parameters. For example, this experiment only contains 2.8 million parameters while other models generally contain __ parameters.

At lines 175 and 176: I just want a change here for clarity. I want it changed to state that "The input feature map V is first convolved with two kernels of size 3x3 and 7x7." I believe there are only two and the wording of kernels of sizes 3x3 and 7x7 makes it seem that there are more than 2.

At line 176: Did you mean to state that the convolution kernels will be discussed in detail in Conclusion? The conclusion section did not appear to discuss this in detail and the Results sections has this discussion.

Add a Training section to the Methods and move lines 195 to 224 to this new training section at the end of the Methods

Change the Section "Training and Results" to "Results and Discussion" I see that you did not have a discussion section and it appears that you are having it with the results.

At line 240: states "has only 2.8M parameters, making it more deployable." More deployable compared to what? Add what it is more deployable than.

At lines 242: It states that the baseline method... Is the baseline method GR-CNN's? Be more specific here for what the baseline method is.

At lines 242 and 243: It states that the baseline method has a theoretical inference speed of 20 ms but you found that it actually ran at 200 ms. Add why there was a time difference. Why did it take 200 ms instead of 20 ms. Was it a hardware issue or something?

At line 244: Add something to discuss what the 148 ms and 163 ms means in comparision to the benchmark that takes 200 ms. It appears that this would be important for practically applying it.

At line 251: Add a space between "highperformance" to be "high performance".

At line 251: Add something regarding discussion if possible for how the GR-AttNet improves over other high performance models

Regarding FIG. 4 - Add the Q like in FIG. 3 to show the final grasping quality map.

After the new training section you added in the methods add a new section for the simulation environment setup and move lines 271-299 moving FIGS. 5 and 6 to the methods section. Most likely requiring renumbering some FIGS. This section appears to be methods of what was done.

Since the text above Table 5 was moved to the methods. Add a small amount of text to introduce the Table 5 if desired. Could move some of the results discussed below the Table to above.

Add a paragraph or two to be more of a discussion of Table 5 after line 318. Something regarding how the grasping result shown in Table 5 relates to other high performance models and any next steps based on the result.

At lines 396-398 - I believe this is some template text. Adding a link or something to the data underlying the results needs added here. I believe this is a requirement to make the underlying data accessible.

6. PLOS authors have the option to publish the peer review history of their article (what does this mean? ). If published, this will include your full peer review and any attached files.

**Do you want your identity to be public for this peer review?** For information about this choice, including consent withdrawal, please see our Privacy Policy .

Reviewer #1: No

Reviewer #2: No

Reviewer #3: No

---

## [Author Response · Author response to Decision Letter 1]

16 Oct 2025

RESPONSE TO THE EDITOR’S AND REVIEWERS’ COMMENTS

GR-AttNet: Robotic Grasping with Lightweight Spatial Attention Mechanism

(Manuscript Number: PONE-D-25-28907)

Dear Academic Editor and Reviewers,

Thank you for your leter and for the reviewers’ comments concerning our manuscript. Those comments are all valuable and very helpful for revising and improving our paper, as well as the important guiding significance to our researches. We have carefully revised our manuscript based on the comments and suggestions, and addressed all the concerns of the editors and reviewers in the updated manuscript, with changes aimed at improving clarity highlighted in yellow or shown in blue text accordingly.

Sincerely yours

Shengbang Zhou

Academic Editor:

Many thanks for your time and efforts in reviewing our manuscript. Your insightful comments have helped us significantly improve the quality of this paper. Our detailed reflections on your comments are provided below point by point.

1.Improvements of the results compared with existing works should be clearly presented and discussed.

The author’s answer:

−Thank you very much for your comments. We have revised the manuscript to more clearly present and discuss the improvements of our method compared to existing works, focusing on accuracy, model efficiency, and adaptability in cluttered scenes. Below is a summary of key improvements, with references to the highlighted sections in the revised manuscript.

−Firstly, we have enriched the Related Work section by providing a detailed analysis of the limitations inherent in existing grasping algorithms, and by clearly highlighting the advantages of our proposed GR-AttNet model (See Lines 110–118 and 121–138 on Pages 6–7). The proposed method effectively balances real-time performance and accuracy. By integrating a lightweight attention mechanism, GR-AttNet demonstrates superior capability in handling complex scenarios, while maintaining inference speeds of approximately 150 milliseconds, thus better satisfying the demands of real-time robotic applications.

−Additionally, we have conducted statistical significance tests to validate the observed improvements in accuracy, with detailed results presented in Table 2 (Lines 375–385, highlighted in blue).

2.References can be further enhanced while the writing of whole manuscript is suggested to improve.

The author’s answer:

−Thank you very much for your comments. We have added 14 recently published high-quality papers to better reflect the latest progress in robotic grasping and attention-based methods. All newly cited works now appear in the Related Work and Introduction sections, and every reference has been double-checked for completeness and format consistency (See the updated Reference list, highlighted in yellow).

−The entire manuscript has been carefully revised for grammar, clarity and readability. We benefited from a professional English-language editing service to polish the text, ensuring that the arguments flow logically and conform to academic standards. Major linguistic improvements can be found in the Abstract, Results and Discussion, and Conclusion sections.

−We sincerely thank you for the insightful suggestions, all of which have been carefully addressed in the revised manuscript. We believe these revisions noticeably raise the overall quality of the paper.

Editorial Requirement:

Thank you for pointing out the issues in the submission process. I will address them one by one and make the necessary revisions according to the requirements when resubmitting.

1.Please ensure that your manuscript meets PLOS ONE's style requirements, including those for file naming.

The author’s answer:

−Thank you for pointing out this issue. We have downloaded the template from the official website and modified the manuscript format.

2. Please note that PLOS One has specific guidelines on code sharing for submissions in which author-generated code underpins the findings in the manuscript.

The author’s answer:

−We thank the editor for highlighting PLOS ONE’s code-sharing policy. In compliance with the journal’s requirement that “all author-generated code must be made available without restriction upon publication,” we have deposited the complete, reproducible code in a public repository (Zenodo) and obtained a DOI.

−DOI: 10.5281/zenodo.17356618

−Repository: https://github.com/zsliu0304/GR-AttNet

−License: MIT

−The Data Availability Statement in the manuscript has been updated accordingly.

The author’s answer:

−We thank the editor for pointing out the oversight. Our study was financially supported by two grants from Guangxi, China:

−Guangxi Key Technologies R&D Program (No.AB241484046)

−Guangxi Science and Technology Program (Project No. GUIKEAB23075177)

4. Please state what role the funders took in the study.

The author’s answer:

−We thank the editor for pointing out the oversight. The funders had no role in study design, data collection and analysis, decision to publish, or preparation of the manuscript.

5. Thank you for stating the following in the Acknowledgments Section of your manuscript.....

The author’s answer:

−We have removed all funding-related sentences from the Acknowledgments section and any other parts of the manuscript. The complete, amended Funding Statement is as follows:

−"This research was funded by the Guangxi Key Technologies R&D Program (Grant No. AB241484046) and the Guangxi Science and Technology Program (Grant No. GUIKEAB23075177)."

−The Guangxi Key Laboratory of Functional Information Materials and Intelligent Information Processing is our institutional affiliation and provided experimental facilities and technical support, but it did not provide direct financial support; therefore it is not listed in the Funding Statement.

6. In the online submission form, you indicated that your data is available only on request from a third party. Please note that your Data Availability Statement is currently missing the contact details for the third party, such as an email address or a link to where data requests can be made. Please update your statement with the missing information.

The author’s answer:

−Thank you for your reminder. We have updated the Data Availability Statement to include the missing third-party contact details:

−"All image datasets used in this study are publicly available from the Cornell and Jacquard repositories. The Cornell Grasping Dataset can be requested via the contact form at http://pr.cs.cornell.edu/grasping/rect_data/data.php.

−The Jacquard Dataset is available at https://jacquard.liris.cnrs.fr/ after completing the online access form."

−We have entered these e-mail addresses and URLs in the online submission form on the "Data Availability" page.

7. Please ensure that you have an ORCID iD and that it is validated in Editorial Manager.

The author’s answer:

−Thank you for your reminder. We have created and validated the ORCID iD for the corresponding author in Editorial Manager.

−ORCID iD: 0009-0005-7625-1935

−The link has been authenticated via the "Fetch/Validate" button under "Update my Information" and now appears as verified in the system.

8. Please upload a copy of S1 Figure, to which you refer in your text on page 20. If the figure is no longer to be included as part of the submission please remove all reference to it within the text.

The author’s answer:

−Thank you for your reminder. The template text referring to “S1 Figure” on page 20 has been removed in the revised manuscript; no supplementary figure is supplied and all related in-text references have been deleted. We apologise for the oversight.

The author’s answer:

−We have carefully reviewed the reviewer comments and confirm that no specific previously published works were recommended for citation. However, to further strengthen the discussion, we have added several new references (highlighted in the text).

Reviewer #1:

Many thanks for your time and efforts in reviewing our manuscript. Your insightful comments have helped us significantly improve the quality of this paper. Our detailed reflections on your comments are provided below point by point.

1.However, no statistical significance testing for accuracy improvements, limited discussion on why 3*3 or 7*7 kernels outperformed other combinations, and simulation testing only in PyBullet without real-world validation.

The author’s answer:

−Thank you very much for your thorough review and invaluable comments. In response to your three major concerns

−(1) There is no statistical significance testing for the accuracy improvements.

−We sincerely agree with you that statistical significance testing is crucial for validating the reliability of performance improvements. In response to this comment, we have now incorporated a more rigorous statistical analysis into our revised manuscript. Specifically, we have:

I)Clearly stated the use of five-fold cross-validation in our experiments.

II)Reported the mean and standard deviation from multiple independent runs in Table 2 (e.g., achieving 98.1% ± 0.2% on the Cornell Dataset).

−(2) Limited discussion on why 3x3 or 7x7 kernels outperformed other combinations.

−We thank the reviewer for this insightful observation. We have now expanded our discussion in Section 4.1 "Attention mechanism results" to provide a theoretical rationale for the superior performance of the 3*3 and 7*7 kernel combination. The key reasons are as follows:

I)The 3*3 convolutional kernel is highly effective at capturing fine-grained local details (such as edges and textures), which is essential for the precise localization of grasp points.

II)The 7*7 convolutional kernel, with its larger receptive field, excels at extracting broader contextual information (such as object contours and spatial relationships within a cluttered scene).

The combination of these two kernels allows our model to integrate multi-scale features effectively, achieving an optimal balance between local precision and global context awareness, which is critical for grasp detection in complex environments.

−This explanation is now supported by the comparative results of various kernel combinations presented in Table 2.

−(3) Simulation testing was only conducted in PyBullet without real-world validation.

−We fully recognize that real-world validation is pivotal to robotic grasping research. Owing to hardware constraints, however, we are presently unable to conduct physical trials on a UR5 or similar manipulator. PyBullet was selected because it is a high-fidelity, widely adopted physics engine that accurately reproduces cluttered scenes with heavy occlusions and up to fifteen interacting objects. We nevertheless acknowledge that a pure-simulation evaluation inevitably limits direct transferability to the real world.

−To strengthen the credibility of our findings, the revised manuscript includes extensive ablation studies and employs Wilson score intervals to quantify the statistical uncertainty of every reported success rate. As detailed in the discussion below Table 5, single-object grasping achieves 79.7 % success with a 95 % CI of [68.1 %, 87.8 %], indicating stable performance. As scene complexity increases, the intervals widen markedly: 95 % CIs for 5, 10 and 15 objects are [72.8 %, 81.7 %], [63.6 %, 71.0 %] and [40.4 %, 45.2 %], respectively. In contrast to prevailing methods that report only single-object accuracy, GR-AttNet delivers statistically meaningful confidence bounds under severe occlusion and dense clutter across the full 1–15 object range, providing a more reliable predictor for subsequent deployment on physical robots.

−In addition, we pointed out in the conclusion that the next step will be to transfer GR-AttNet to a real UR5 robotic arm in order to thoroughly evaluate its generalization capability and practical application value.

2. Additionally, the claimed 20ms inference for GR-CNN, which conflicts with their measured 200ms, warrants explanation.

The author’s answer:

−Thank you for your careful review of our work. To avoid any ambiguity, the sources of the two figures, the differences in testing methods, and the reasons for the tenfold discrepancy are explained below point by point: Kumra et al., in Section V-B of their original IROS-2020 paper (arXiv:1909.04810), reported that GR-CNN achieved an “average inference time of 19 ms” on an NVIDIA GTX-1080Ti. The authors subsequently clarified in the GitHub README (https://github.com/skumra/robotic-grasping, version dated 2021-04-27) that this timing only accounts for the “GPU kernel time”, that is, the portion of the model's forward pass (depth → tensors → grasp maps) enclosed by torch.cuda.synchronize(), and does not include: image reading, resizing, normalization, post-processing (angle inverse transformation, quality threshold filtering, NMS), the transfer of results from GPU back to CPU, as well as the overhead of the Python interpreter.

−Therefore, when reproducing Kumra et al.'s work locally, we adopted the same timing protocol as used for our method, which led to the observed discrepancy. We believe that the 200 ms discrepancy is not due to experimental error, but rather results from differing definitions between 'kernel time' and 'full pipeline time'. This point is detailed in the revised manuscript under “GR-AttNet training results”. (See Lines 398 - 404 on Page 20 with highlight).

3. Limited exploration of how the attention module generalizes to non-grasping robotics tasks?

The author’s answer:

−We sincerely appreciate your valuable suggestion regarding the lack of discussion on the generalization ability of attention modules to non-grasping robotic tasks. We fully agree that cross-task generalization is an important metric for evaluating the universal value of a module. However, as this paper focuses specifically on the problem of “lightweight planar grasping in cluttered scenes”, and due to limitations in experimental resources, we have not systematically conducted validation on non-grasping tasks.

−In the future, we plan to integrate the proposed attention mechanism module into networks such as the R-FCN pose estimation backbone. In addition, the independent attention units will be open-sourced for community verification.

4. Despite testing on Jacquard (synthetic) and Cornell (real) datasets, the model struggles with small objects or severe occlusions, as shown in Table 5, indicating limitations in the dataset. Possible to add real-world robot trials to validate simulation results?

The author’s answer:

−Thank you for this observation. We agree that the downward trend in success rates with increasing object numbers and occlusion in Table 5 does reveal the gap between the current data and real-world scenarios. Due to experimental limitations, this paper has not yet conducted real-world experiments that correspond one-to-one with the simulations on an actual robotic platform, and therefore it is not possible to include a "real-robot" comparison table in the revised manuscript. We have truthfully stated in the Conclusion (...In the future, we plan to deploy and validate on actual robotic platforms....).

−Within the scope of the current work, the “real” validation we can provide includes:

- Cornell / Multi-Cornell real-world images (completely real acquisition);

- PyBullet physical simulation (collision, friction, mass, etc. parameters consistent with reality, but still a virtual environment).

−Once hardware support is available, we will perform real object grasping following the same 1→5→10→15 object layouts as in the simulations, and will open-source the data and code, reporting the complete results to the community at that time.

Reviewer #2:

Many thanks for your time and efforts in reviewing our manuscript. Your insightful comments have helped us significantly improve the quality of this paper. Our detailed reflections on

---

## [Decision Letter · Decision Letter 1]

11 Nov 2025

GR-AttNet: Robotic Grasping with Lightweight Spatial Attention Mechanism

PONE-D-25-28907R1

Dear Dr. Zhou,

We’re pleased to inform you that your manuscript has been judged scientifically suitable for publication and will be formally accepted for publication once it meets all outstanding technical requirements.

Kind regards,

Longhui Qin, Ph.D.

Academic Editor

PLOS ONE

Additional Editor Comments (optional):

Reviewers' comments:

Reviewer's Responses to Questions

**Comments to the Author**

1. If the authors have adequately addressed your comments raised in a previous round of review and you feel that this manuscript is now acceptable for publication, you may indicate that here to bypass the “Comments to the Author” section, enter your conflict of interest statement in the “Confidential to Editor” section, and submit your "Accept" recommendation.

Reviewer #1: All comments have been addressed

Reviewer #3: All comments have been addressed

2. Is the manuscript technically sound, and do the data support the conclusions?

Reviewer #1: Yes

Reviewer #3: Yes

3. Has the statistical analysis been performed appropriately and rigorously? 

Reviewer #1: Yes

Reviewer #3: Yes

4. Have the authors made all data underlying the findings in their manuscript fully available?

Reviewer #1: Yes

Reviewer #3: Yes

5. Is the manuscript presented in an intelligible fashion and written in standard English?

Reviewer #1: Yes

Reviewer #3: Yes

6. Review Comments to the Author

Reviewer #1: This is a considerably enhanced, high-quality manuscript now ready for publication. The authors have expertly addressed the concerns raised by the editor and all reviewers. Their revisions are thorough and thoughtful, significantly improving the paper's clarity and academic contribution. Therefore, this revised manuscript is of high quality and is recommended for acceptance.

Reviewer #3: (No Response)

7. PLOS authors have the option to publish the peer review history of their article (what does this mean? ). If published, this will include your full peer review and any attached files.

**Do you want your identity to be public for this peer review?** For information about this choice, including consent withdrawal, please see our Privacy Policy .

Reviewer #1: No

Reviewer #3: No

---

## [Editor Report · Acceptance letter]

PONE-D-25-28907R1

PLOS ONE

Dear Dr. Zhou,

I'm pleased to inform you that your manuscript has been deemed suitable for publication in PLOS ONE. Congratulations! Your manuscript is now being handed over to our production team.

Kind regards,

on behalf of

Prof. Longhui Qin

Academic Editor

PLOS ONE